# Cooling lactating sows exposed to early summer heat wave alters circadian patterns of behavior and rhythms of respiration, rectal temperature, and saliva melatonin

Wonders Ogundare[1], Kelsey Teeple[2], Elizabeth Fisher[2], Corrin Davis[2], Leriana Garcia Reis[2], Amber Jannasch[3], Linda M. Beckett[2], Allan Schinckel[2], Radiah Minor[1], Theresa Casey[2]*

1 Department of Animal Sciences, North Carolina Agricultural and Technical State University, Greensboro, North Carolina, United States of America, 2 Department of Animal Sciences, Purdue University, West Lafayette, Indiana, United States of America, 3 Metabolite Profiling Facility, Bindley Bioscience Center, Purdue University, West Lafayette, IN, United States of America

* theresa-casey@purdue.edu

**Data Availability Statement:** All relevant data are within the paper and its Supporting Information files.

## Abstract

Heat stress (HS) exerts detrimental effects on animal production, with lactating sows being particularly vulnerable. Understanding the mechanisms involved in HS response could aid in developing effective strategies against the negative impacts on livestock. Recent genome wide association studies identified two core circadian clock genes as potential candidates in mediating HS response. The study aimed to investigate how cooling lactating sows under natural heat stress conditions impacted circadian patterns of respiration rate (RR), rectal temperature (RT), behavior, salivary melatonin and cortisol levels, and diurnal patterns of cytokines in saliva. Mixed parity lactating sows were assigned to one of two treatment groups: electronic cooling pad (C; n = 9) and heat-stressed (H; n = 9). The experiment spanned two 48 h periods of elevated ambient temperatures due to summer heat wave. In the first 48 h period, RR was recorded every 30 min, RT every 60 min, and behaviors (eating, standing, sitting, laying, sleeping, drinking, and nursing) every 5 min. In the second 48 h period, saliva samples were collected every 4 h. Cooling reduced RR and RT and altered circadian patterns (P < 0.05). Cooling did not affect amount of time engaged in any behavior over the 48 h period (P > 0.05), however, daily patterns of eating, standing and laying differed between the treatments (P < 0.05), with altered eating behavior related to RT increment in H sows (P < 0.05). Cooling increased and altered the circadian pattern of salivary melatonin (P < 0.05). Cooling also influenced the diurnal pattern of saliva cytokines. Cooling had no impact on saliva cortisol levels. In conclusion, cooling HS sows impacted circadian rhythms of physiology and behavior, supporting the need for further research to understand if circadian disruption underlies decreased production efficiency of HS animals.

**Funding:** This research was supported by the U.S. Department of Agriculture, National Institute of Food and Agriculture (USDA-NIFA) grant 2021-07097 The impact of moringa on milk quality and quantity and piglet health in heat stressed swine.

**Competing interests:** The authors have declared that no competing interests exist.

## Introduction

Globally swine production is a flourishing industry, with pork one of the most consumed meats worldwide. However, heat stress (HS) poses significant challenges to the swine industry, resulting in a yearly economic loss of approximately $1 billion for United States pork producers [1]. Lactating sows are particularly vulnerable to HS due to the intense metabolic activities associated with milk production, with thermoneutral zone of lactating sows ranging from 18–22˚C compared to 22–25˚C in growing pigs [2]. Internal heat load from lactation coupled with high ambient temperature reduces feed intake, which negatively impacts milk production and consequently impairs suckling litter growth [2, 3]. High heat load also leads to increased respiration rate (RR), rectal temperature (RT), and changes in feeding, drinking, standing, and sitting behaviors in sows during late lactation [4]. Alterations in the immune system occur in HS animals, and may be due in part to changes in blood distribution and intestinal barrier function, which potentiate systemic inflammation [5–7]. In swine, lower litter growth rate is primarily attributed to lower milk production due to decreased feed intake of the heat stressed sow [8]. However, lower feed intake alone cannot fully account for the poor growth rates of litters [9]. Understanding the mechanisms involved in the HS response, and relating these to production performance may aid in the development of effective strategies in mitigating the negative impacts on swine litter growth.

Recent genome and phenome-wide association studies of rats, cattle and humans found five genes related to the HS response common across these mammals. Of the five genes, two were core circadian clock genes, basic helix-loop-helix ARNT like 1 (ARNTL) and neuronal PAS domain protein 2 (NPAS2) [10], which suggests that the circadian timing system is altered during the HS response and may underlie variations in heat tolerance. Core clocks generate circadian rhythms to coordinate internal physiology and synchronize it to the external environment [11, 12]. The light-dark cycle is the primary environmental input for entraining the timing of master circadian clocks in the suprachiasmatic nucleus (SCN) while other inputs include behavior and stress [13, 14]. The SCN integrates this temporal information and generates circadian rhythms of body temperature, and hormones like melatonin and cortisol. The circadian clocks are integrated and reciprocally regulated by nearly all systems of the body such as metabolic, reproductive, and immune system [15], and play central roles in mediating homeostatic feedback loops and cellular repair pathways. Input factors to the circadian clocks can however disrupt them when applied at inappropriate and unusual timings, and are termed chronodisruptors, [16]. Chronodisruption is linked to metabolic disturbances, lower immunity and poor reproductive performance [15, 17, 18]. Thus, understanding the impact of HS on circadian physiology is important to understanding how it affects animal production and welfare.

Cooling sows when ambient temperature and humidity are high is an effective means of reducing HS [19, 20]. Recently developed electronic cooling pads consist of an aluminum top-plate with an inlet and outlet valve for temperature regulated water flow that animals lie on to remove heat through conductive cooling. The cooling pads showed good efficacy in reducing RR and maintaining a narrow and lower range of body temperature in lactating sows [8, 21]. While previous studies have explored circadian disruptions in heat-stressed cattle and found changes in behavioral patterns of feeding [22] and laying [23] in dairy cows under heat stress conditions, little is known about the effect of HS on circadian patterns and the potential mitigating role of cooling pads in regards to behaviors, hormones and circadian rhythms of physiology especially in lactating sows. Herein, we describe findings regarding the effect of cooling lactating sows exposed to a natural summer heat wave on circadian patterns of RR and RT, saliva concentrations of melatonin and cortisol, behaviors of eating, sleeping, drinking, standing, sitting, nursing, and laying, as well as diurnal levels of saliva cytokines. We hypothesized

that cooling heat-stressed lactating sows would impact circadian patterns of behavior, RR, RT, and saliva hormone levels of melatonin and cortisol, accompanied by diurnal changes in levels of inflammatory cytokines.

## Materials and methods

### Ethics approval

This study was performed at the Swine Unit of Purdue University Animal Research and Education Center (ASREC) following review and approval of the study protocol by the Institutional Animal Care and Use Committee (Purdue IACUC Protocol #2110002202).

### Animals and project overview

The priori sample size calculations indicated we need 12 animals in each treatment group but multiple sows were lost due to diseases unrelated to the study. Regardless, we decided to continue on with the project due to its highly exploratory nature. Power analysis conducted indicated that with 10 animals per treatment, and a 1.5-fold difference in variable with 0.5 standard deviation, the power of the study is 0.80; if the difference increased 1.75, then a sample size of 4 was needed for a power of 0.8. With n = 9 animals in each group, and a 1.75 difference, with 0.5 standard deviation, then the power of the study is 0.9.

A mixture of first parity (P1; n = 11) and second parity (P2; n = 7) Large White-Landrace crossbred sows bred to Duroc terminal sires were used. All sows (n = 18) and their piglets (13 to 14 per sow) were equally divided into two identical farrowing rooms A and B maintained under the same conditions. Sows were fed a corn-soybean based lactation diet (Table 1), analyzed for composition by the Experiment Station Chemical Laboratories at the University of Missouri. The diet met or exceeded NRC 2012 recommendations and was fed for *ad libitum* intake along with *ad libitum* access to water. The rooms were exposed to naturally-occurring summer environmental heat and humidity in June 2022, and sows were equally divided into two treatment groups of cooled (C; n = 9) and HS (H; n = 9). Cooling was achieved by conductive cooling pads (Innovative Heating Technologies; Oak Bluff, MB, Canada) placed on the farrowing crate floor. Conductive cooling was achieved by flushing cool water through the pad when the temperature reached 26°C [8, 24]. Room A housed 5 H and 4 C sows, and room B housed 4 H and 5 C sows.

The experiment was a two-phase 48 h observational study performed within two consecutive weeks that had forecasted high environmental temperature and humidity (Fig 1). The light-dark cycle of electric lighting in the room was 16 h light and 8 h dark. The room light was turned on at 0600 and turned off at 2200, making the electric light-dark cycle slightly lengthened relative to the time of natural light exposure with time of sunrise and sunset at 0617 and 2119, respectively. During the dark phase of the light-dark cycle (2200–0600), 250-Watt red light heat lamps (Savant Technologies LLC, Ohio; Producer's Pride, TN; and Feit Electric CA, USA) remained on to maintain piglet temperature.

The first phase, which consisted of behavior observation and RR and RT recording, was a 48 h period that began on June 14, 2022 at 0600. Time destination for the first phase ran from 0600 and continued for 48 h, with final data point collected at 5400, which was 0600 on June 16, 2022.

### Respiration rate and rectal temperature data collection

Daily rhythms of RR and RT were captured during the first phase of the study, which encompassed day 4±3 of lactation across all sows. RR and RT were recorded every 30 min and 60

**Table 1. Lactation diet composition (as-fed basis).**

| Item | Lactation |
|---|---|
| *Ingredient, %* | |
| Corn | 58.75 |
| Soybean meal | 34.00 |
| Moringa dried leaf powder | 0.00 |
| Swine grease | 3.00 |
| Limestone | 1.43 |
| Monocalcium phosphate | 1.34 |
| Phytase | 0.10 |
| Provimi Sow Vitamin+TM Premix | 0.15 |
| Choline Cloride (60%) | 0.10 |
| Rovimix-CarniChrom | 0.01 |
| Salt | 0.50 |
| Availa Sow | 0.075 |
| Clarify | 0.10 |
| Defusion Plus / myco prevent | 0.25 |
| Titanium Premix | 0.20 |
| Total | 100.00 |
| *Calculated analysis* | |
| Metabolizable Energy, Kcal/kg | 3359 |
| Crude Protein, % | 21.08 |
| Total Lysine, % | 1.15 |
| SID Lysine, % | 1.00 |
| Calcium, % | 0.90 |
| Total Phosphorus, % | 0.68 |
| ATTD Phosphorus, % | 0.45 |
| *Analyzed composition* | |
| Ash, % | 6.82 |
| Crude Protein, % | 20.54 |
| Crude Fiber, % | 2.71 |
| Crude Fat, % | 5.48 |
| Moisture, % | 4.54 |
| Lysine, % | 1.17 |
| Methionine, % | 0.31 |
| Cysteine, % | 0.32 |
| Threonine, % | 0.78 |
| Tryptophan, % | 0.30 |
| Isoleucine, % | 0.94 |
| Leucine, % | 1.82 |
| Valine, % | 1.02 |

min, respectively, over the 48 h period. One breath was measured as one rise and fall of the flank or frequency of nose crunches within a minute, calculated by using a stopwatch for 15 sec and count multiplied by 4. RT was measured using a digital thermometer (Cooper Atkins, TM99A-V, USA) with the probe inserted at least 6 cm into the rectum of the sow and left until a stabilized reading was observed and recorded.

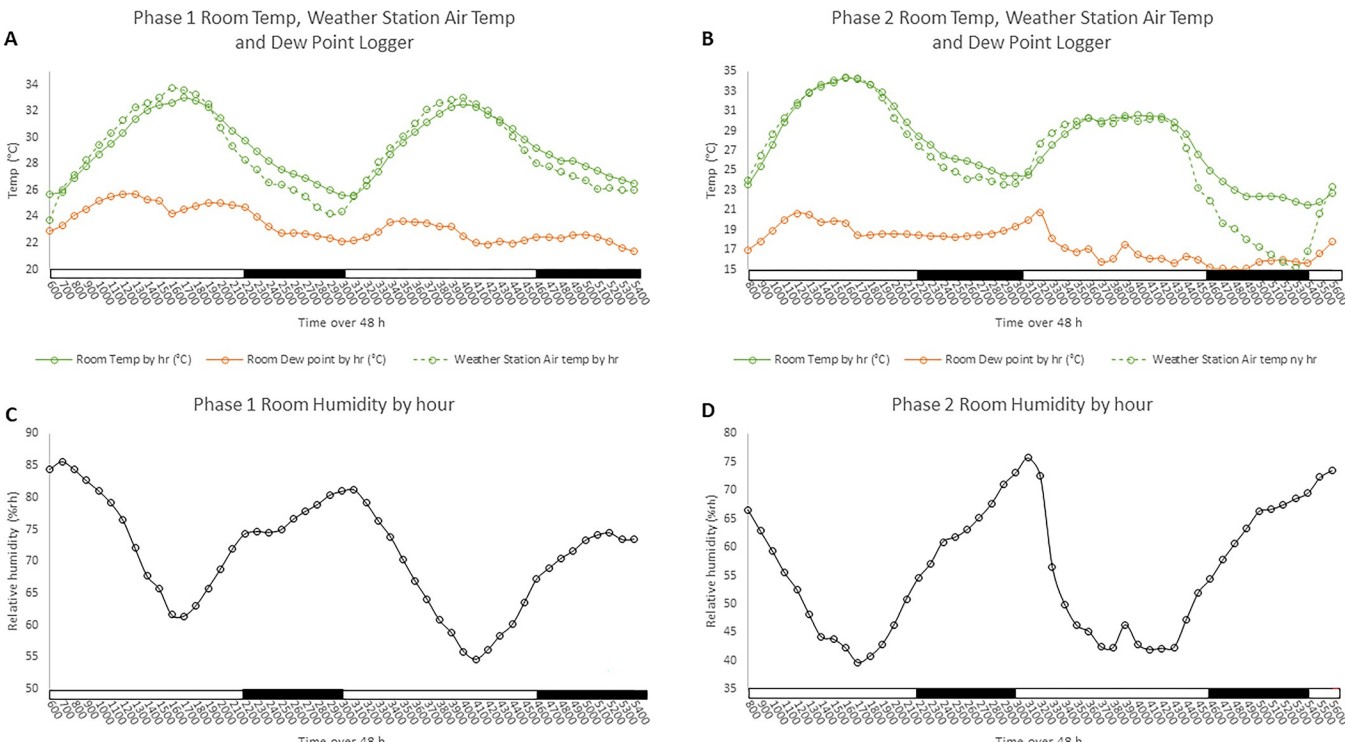

**Fig 1. Temperature, dew point, and relative humidity during experiments.** Experimental phase 1 (A) and phase 2 (B) hourly room temperature (solid green line) and dew point (orange line) recorded by temperature loggers in experimental room, and air temp (dotted green line) as recorded by the weather station in West Lafayette, Indiana. Phase 1 (C) and phase 2 (D) hourly relative humidity (% rh) recorded by temperature loggers. Black bars on the x-axis represent the dark phase while the white bars represent the light phase of the 16 h light and 8 h dark cycle.

## Behavior data collection

Behavior data collection was taken simultaneously with RT and RR data during the first phase of the experiment. The frequency of behavior data collection was every 5 min. The sows were observed for eating, laying, standing, sitting, drinking, sleeping, and nursing, while the piglets were observed for sleeping, nursing, and walking. Sleeping was defined as inactivity, lying with eyes closed. Sows were noted to be nursing when all-to-most piglets were actively on the teats (nutritive suckling) as opposed to non-nutritive suckling with only one or two piglet(s) suckling. Behavior was recorded while being as non-invasive as possible, to include minimal movement and noise by researchers, so as not to disrupt sow and piglet behavior or interfere with data collection, with the acknowledged potential that RT measures may have disrupted behavior.

## Behavior data analysis

Each behavior data was summed every hour by taking 5 min observation as one (1) count. Thus, an hour of full activity for each behavior was 12 counts. The percentage of time in performing each activity was calculated by dividing the total counts in an hour by 12 and multiplying by 100. This was calculated for the first and the second 24 h period.

## Saliva sampling

During the second phase of the circadian study, saliva was collected every 4 h for 48 h from all sows (n = 18) starting at 0800 on post farrowing d 12±4. Saliva was collected using salivettes

(Sarstedt, Germany) secured with hemostats to place and hold in the mouth of the sows for about 2 min. The salivette was centrifuged at 1000 × g for 15 min to extract the saliva and pellet cellular debris. Saliva was aliquoted into multiple tubes, and temporarily kept at -20°C on the farm then transferred to -80°C at the laboratory.

## Cortisol and melatonin measurement

Saliva cortisol and melatonin were measured using liquid chromatography tandem mass spectrometry (LC-MS/MS) by adapting a protocol from [25]. The extraction and analysis of samples was done in the Metabolite Profiling Facility of Bindley Biosciences Center at Purdue University. To 100 μL of saliva sample, 10 μL of internal standard (ISTD; 0.5ng deuterium cortisol-d4/melatonin-d4; LGC Limited, Middlesex, UK; Cayman Chemical, MI, USA) along with 500 μL of Methyl tert-Butyl Ether (MtBE) were added. If less than 100 μL of a saliva sample was available, volume was recorded, and downstream calculations were modified to account for difference. The mixture was vortexed for 30 sec and centrifuged at 16,000 x g for 10 min at room temperature. The organic top layer was collected and dried in a speed vacuum for 3 h or overnight at room temperature. Dried samples were temporarily stored at -20°C until further analysis.

The experimental procedure followed with the reconstitution of dried samples in a 50 μL solution composed of 20% methanol and 80% water. Subsequently, LC-MS/MS was employed for the precise quantification of cortisol and melatonin concentrations within each sample. The analytical instrumentation was an Agilent 1290 Infinity II liquid chromatography (LC) system coupled with an Agilent 6470 series triple quadrupole mass spectrometer (QQQ-MS/MS) (Agilent Technologies, Santa Clara, CA). Separation during liquid chromatography was accomplished utilizing an Agilent Eclipse plus C18 column (2.1 mm x 50 mm, 1.8 μm). The mobile phase consisted of (A) water with 0.1% formic acid and (B) acetonitrile with 0.1% formic acid. The linear gradient for liquid chromatography was as follows: 0 minutes, 5% B; 1.0 minutes, 5% B; 10.5 minutes, 65% B; 11 minutes, 98% B; 11.2 minutes, 5% B; 14 minutes, 5% B. The flow rate was consistently maintained at 0.3 mL/min. Elution times for melatonin and cortisol were observed at 5.1 minutes and 6.1 minutes, respectively. Multiple reaction monitoring, under positive electrospray ionization (ESI) mode, was employed for mass spectrometric analysis. The parameters for the ESI interface included a gas temperature of 325°C, a gas flow rate of 7 L/minute, a nebulizer pressure of 45 psi, a sheath gas temperature of 250°C, a sheath gas flow rate of 7 L/minute, a capillary voltage of 4000 V in positive mode, and a nozzle voltage of 1500 V. The ΔEMV voltage was set at 500 V. Data analysis was conducted utilizing Agilent Masshunter Quantitative Analysis software (version 10.1) using internal standards for quantification. For quantitation of melatonin/d4-melatonin, the transition 233.7 → 174.4/237.4 → 178.4 was used. For cortisol/d4-cortisol, the transition 363.5 →121.2/367.5 → 121.2 was used and followed the analysis in Table 2. The concentration of cortisol was determined in ng/mL, computed by multiplying the ratio of cortisol to d4cortisol by the internal standard value of 0.5 ng. Similarly, the concentration of melatonin was calculated using the same method.

## Cytokine measurement by relative fold change in saliva

Saliva cytokine concentrations were analyzed using a Porcine T Cell Response Array 1 (Ray Biotech, GA). A single pool of morning (0800) and evening (2000) samples was prepared for each treatment (H and C). Pools were prepared by aliquoting 20 μL of saliva from each sow at each time point and group (H, n = 7; C, n = 7). Pooling samples resulted in 4 distinct samples: H AM (0800), H PM (2000), C AM (0800), and C PM (2000). Pooled samples were diluted 3-fold with a blocking buffer (250 μL pooled saliva sample to 750 μL blocking buffer) and the

**Table 2. Multiple reaction monitoring for data acquisition of cortisol and melatonin saliva concentrations.**

| Compound Name | Precursor Ion (m/z) | Product Ion (m/z) | Collision Energy (V) |
|---|---|---|---|
| Cortisol | 363.5 | 327.5 | 20 |
| Cortisol | 363.5 | 121.2 | 20 |
| Cortisol-d4 | 367.5 | 331.5 | 20 |
| Cortisol-d4 | 367.5 | 121.2 | 20 |
| Melatonin | 233.4 | 174.4 | 10 |
| Melatonin | 233.4 | 159.3 | 35 |
| Melatonin-d4 | 237.4 | 178.4 | 10 |
| Melatonin-d4 | 237.4 | 163.3 | 35 |

d4 –Deuterium4

V–Velocity

m/z–mass to charge ratio

cytokine array analysis was performed per manufacturer's protocol. Images of array membranes were obtained by chemiluminescence using a ChemiDoc MP Imaging System (Bio Rad Laboratories, CA). Densitometry was utilized for each membrane treatment group using Image Lab 6.0.1 software. Data are represented as a relative fold change of the average measured density of each cytokine to the average density of positive spots on each membrane, and the resultant ratio then divided by the average of each cytokine over the 4 membranes and plotted using Microsoft Excel for Microsoft 365 (Version 2304 Build 16.0.16327.20200).

## Data and statistical analysis

Data analysis was conducted using SAS On Demand for Academics (SAS Institute Inc, Release 3.1.0, NC, USA), employing the PROC MIXED model. The mixed model included the fixed effects of day, time, and treatment with sow considered as a random effect and time treated as a repeated measurement. Statistical significance was determined at $P < 0.05$ and tendencies at $P < 0.10$. Treatment means were compared at each time point using the slice procedure of SAS PROC MIXED. Regression analysis was performed using the treatment effects and time of day continuous variables X1, X2, X3, X4, X5, and X6, which were incorporated into a Fourier series model. The Fourier series model variables X1 and X2 represented the sine and cosine single phases (24 h), X3 and X4 represented the sine and cosine double phases (12 h), and X4 and X6 represented the sine and cosine triple phases (8 h) of shifts in the curve [26, 27]. The regression analyses were completed using the PROC MIXED procedure including the main effects of treatment and day, the periodic regression variables and sow as a random effect. The regression analysis examined the interactions of treatments with the sine and cosine functions. Non-significant interactions with P values greater than the declared P-value ($P < 0.05$) were removed first, followed by the sine-cosine independent variables, except in cortisol and melatonin, where significant tendencies remained at $P < 0.20$. The mean predicted values obtained from the sine and cosine analysis were plotted against the raw data of RR, RT, cortisol and melatonin using Microsoft Excel. Observed behaviors of eating, standing, drinking, sitting, laying, nursing, and sleeping were also analyzed using the PROC MIXED procedure for effect of day, time and treatment on percentage of time the sow was performing each behavior. To investigate the potential influence of heat increment from eating on the unexpected peaks in rectal temperature (RT) during the night phase, we conducted periodic regression analysis. Building on previous findings by Noblet et al. (1993), which indicated that

the heat increment in body temperature due to eating occurs approximately 2 h later, we included the level of eating behavior in the prior 2 h to the Fourier series model for RT.

## Results

### Room temperature conditions

External air temperature captured by the local weather station and recorded experimental room temperature exhibited similar variations across the 48 h of the first phase of the experiment. The peak of room temp in phase 1 (Fig 1A) was 32˚C on both days at 1700 and 4000 (1700 on day 1 and 1600 on day 2), and the lowest temperature recorded was 26˚C on both days at 600 and 3100 (600 on day 1 and 700 on day 2). The lowest ambient room temperature occurred during the dark phase on both days. Dew points in the room peaked at 1300 on day 1 at 25˚C and at 3400 (1000) on day 2 at 23˚C. Relative humidity in phase 1 (Fig 1C) peaked at the second hour of each day with 85% rh and 80% rh on day 1 and 2, respectively, with lower percentages through the rest of the day. In the second 48 h period, experimental phase 2, the highest recorded room temp (Fig 1B) was approximately 35˚C on day 1, and 30˚C on day 2 at 1600 and 4000 (1600 on day 2), respectively. Low temperatures of approximately 24˚C on day 1 and 21˚C on day 2 were observed coincident with the onset of the light phase each day. Relative humidity in phase 2 (Fig 1D) was greatest at 800 with approximately 67% rh on day 1 and 3100 with approximately 75% rh on day 2.

### Cooling effect on RR and RT

The effects of cooling, time, day, as well as interactions between cooling and day, and cooling and time were highly significant for sow respiration ($P < 0.001$). Regression analysis using sine and cosine fitting demonstrated that the RR exhibited a 24 h rhythm ($P < 0.05$), and this rhythm was also influenced by cooling (Fig 2A). The effects of cooling, day, and time were significant ($P < 0.05$) for RT (Fig 2B). Regression analysis using the sine and cosine variables revealed that RT exhibited significant 24, 12, and 8 h rhythms ($P < 0.01$). Treatment interactions were associated with both the cosine rhythms of 24 and 8 h, indicating significant differences in the daily patterns of RT for the two groups ($P < 0.05$). In addition to the 24 h rhythm, evident in trough and peak of RT in the light and dark phase, respectively, the H group exhibited 8 h rhythms in RT corresponding to peaks at 1100, 1900, 2600, 3500, 4300, and 5000. The H group primarily exhibited a 24 h rhythm in body temperature with trough and peak in light and dark phase, respectively (Fig 2B).

### Cooling effect on behaviors

Cooling did not affect percent of time sows spent eating across the 48 h period ($P > 0.05$). However, time of day affected eating behavior and there was an interaction between time and cooling ($P < 0.01$). Time slice analysis revealed a significant cooling by time interaction ($P < 0.05$) at 0600, 1500, and 1600 for eating behavior, and a tendency for a difference, $P < 0.10$, at 2000 across both days (Fig 3A). Cooling did not affect the total percent of time sows were standing across the 48 h period ($P > 0.05$), but time of day significantly affected standing behavior, and there was an interaction for treatment by time, and day by time for sow standing ($P < 0.05$). Time slice analysis revealed treatment by time interactions ($P < 0.05$) at 600, 700 and 900 and tendency for a difference ($P < 0.10$) at 1600, 1900, and 2000 across both days (Fig 3B). There was no effect of treatment on percent of total time of sow laying behavior, but significant interactions were observed for time, treatment by time, and day by time ($P < 0.05$). Using time slice analysis, an interaction between treatment and time ($P < 0.10$)

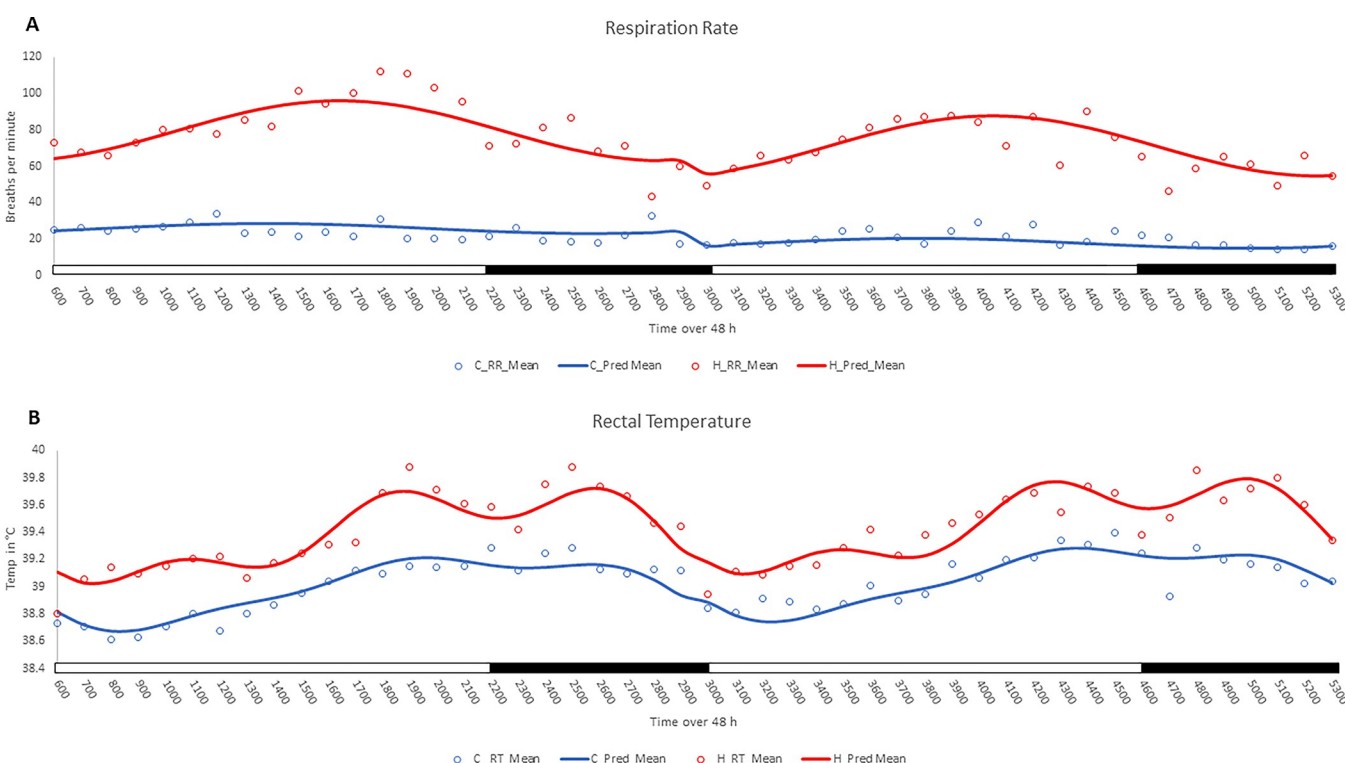

**Fig 2. Effect of Cooling on RR and RT.** The effect of cooling (blue) and H(red) on mean (open circles) RR (A) and RT (B) and the fit to sine and cosine curves (solid line) over 48 h. Black bar on the x-axis represent the dark phase while the white bars represent the light phase.

was evident at 600, 700, 900, 1500, 1600, and 1900 across both days (Fig 3C). The effect of treatment had no influence on percent of time sow spent nursing across the 48 h period, but time of day significantly affected nursing behavior (P < 0.01; (Fig 3D). There was no effect of treatment on sow drinking, sitting, and sleeping behaviors.

Fourier regression models of RT in the H group found two phase (12 h) and three phase (8 h) sine variables, indicating that the second peak in RT during the dark phase of H sows was significant. Peak eating behavior of H animals was later in the day than for C sows (Fig 4). The RT of the H sows had a linear relationship with the percentage of time spent eating the prior 2 h. The RT of H sows spending more time in eating activity in the morning and later in the evening had greater rises in RT above the overall Fourier series predicted values. The relationship between RT and eating behavior indicated the increase in RT later in the evening at 2600 and 5000 in the H sows was caused by the high eating frequency at 2300 and 4600 (Fig 4).

## Cooling effect on cortisol and melatonin concentrations

There was no effect of treatment on salivary cortisol (Fig 5A) over the 48 h period, but cortisol levels varied by time of day (P < 0.05). Sine fit regression analysis indicated that salivary cortisol followed a 24 h rhythm (P < 0.05) and showed a tendency (P < 0.20) to fit sine and cosine rhythms of 24 h, 12 h, and 8 h. The concentration of salivary melatonin (Fig 5B) was significantly affected by effects of cooling, time and day (P < 0.05), and there was a tendency for treatment by time to affect melatonin concentration (P < 0.10). In particular, cooling significantly increased (P < 0.05) salivary melatonin levels across the entire 48 h period, with a trend for higher levels during the light phase of the light-dark cycle. Sine and cosine fit regression analysis revealed that salivary melatonin followed a rhythm every 24, 12, and 8 h (P < 0.05).

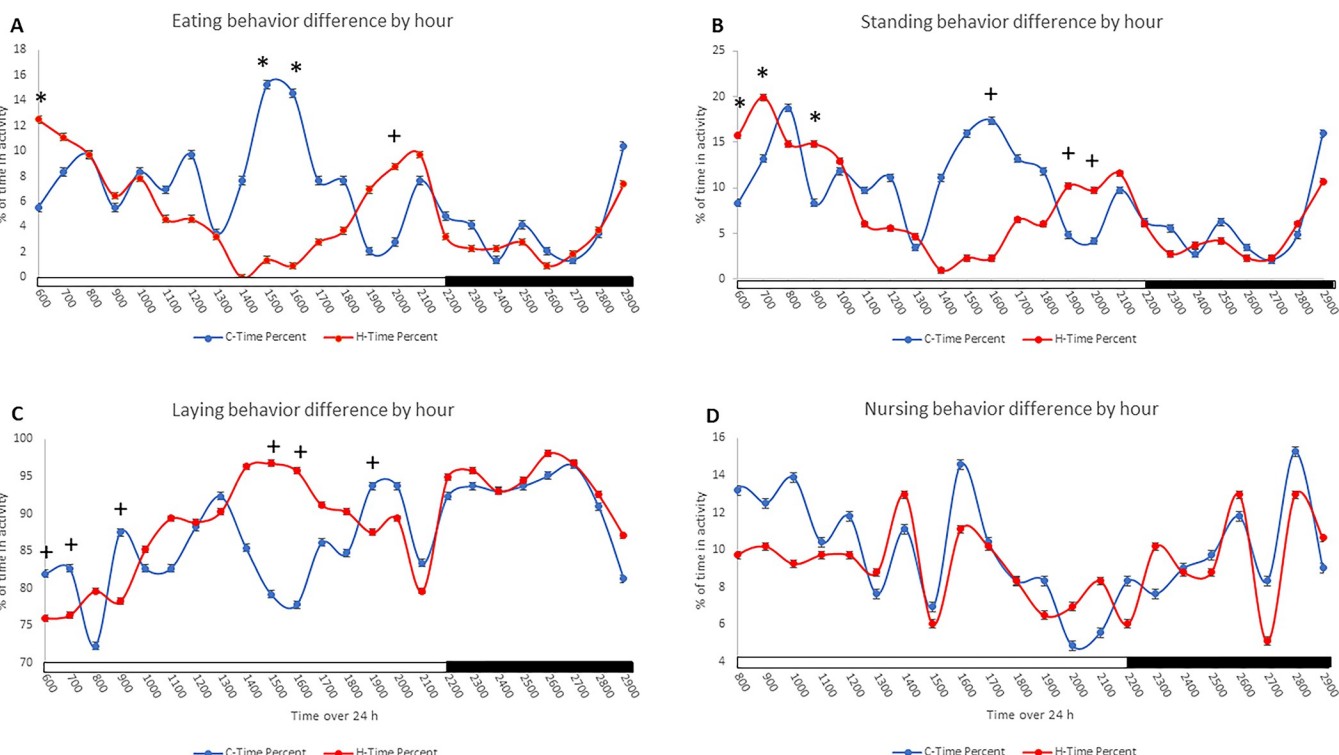

**Fig 3. Cooling on HS sow eating, standing, laying and nursing behaviors.** Effect of cooling (C) (blue) and heat stressed (H) (red) on eating (A), standing (B), laying (C), and nursing (D) behaviors over 24 h from time slice analysis. Black bar on the x-axis represents the dark phase while the white bar represents the light phase. * indicates results from time slice analysis on treatment by time interaction with significance at P < 0.05, and + indicates a significant tendency at P < 0.1.

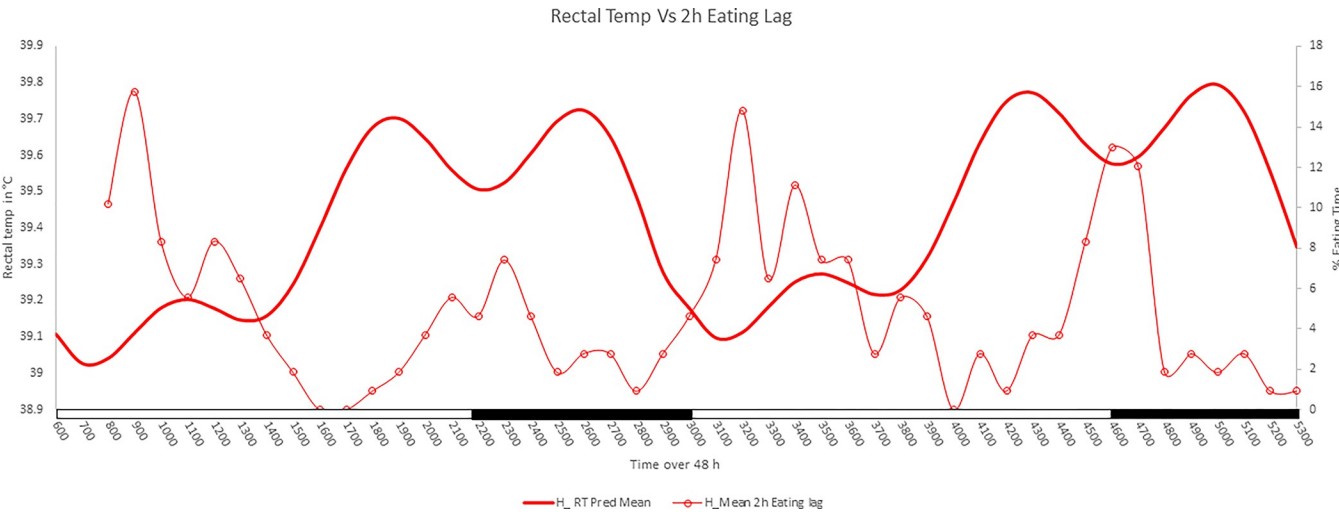

**Fig 4. The relationship between eating behavior to RT increment in HS sows.** The rectal temperature (RT) were fitted to the Fourier series and eating in the previous 2 h included as a linear covariate. The RT (bold smooth line) and a 2 h lag in the time of eating (circled line) of heat-stressed group (H) over a 48 h period indicate a second peak of RT in dark phase, and corresponds to the increased eating 2 h prior. The x-axis has a black bar representing the dark phase and a white bar representing the light phase.

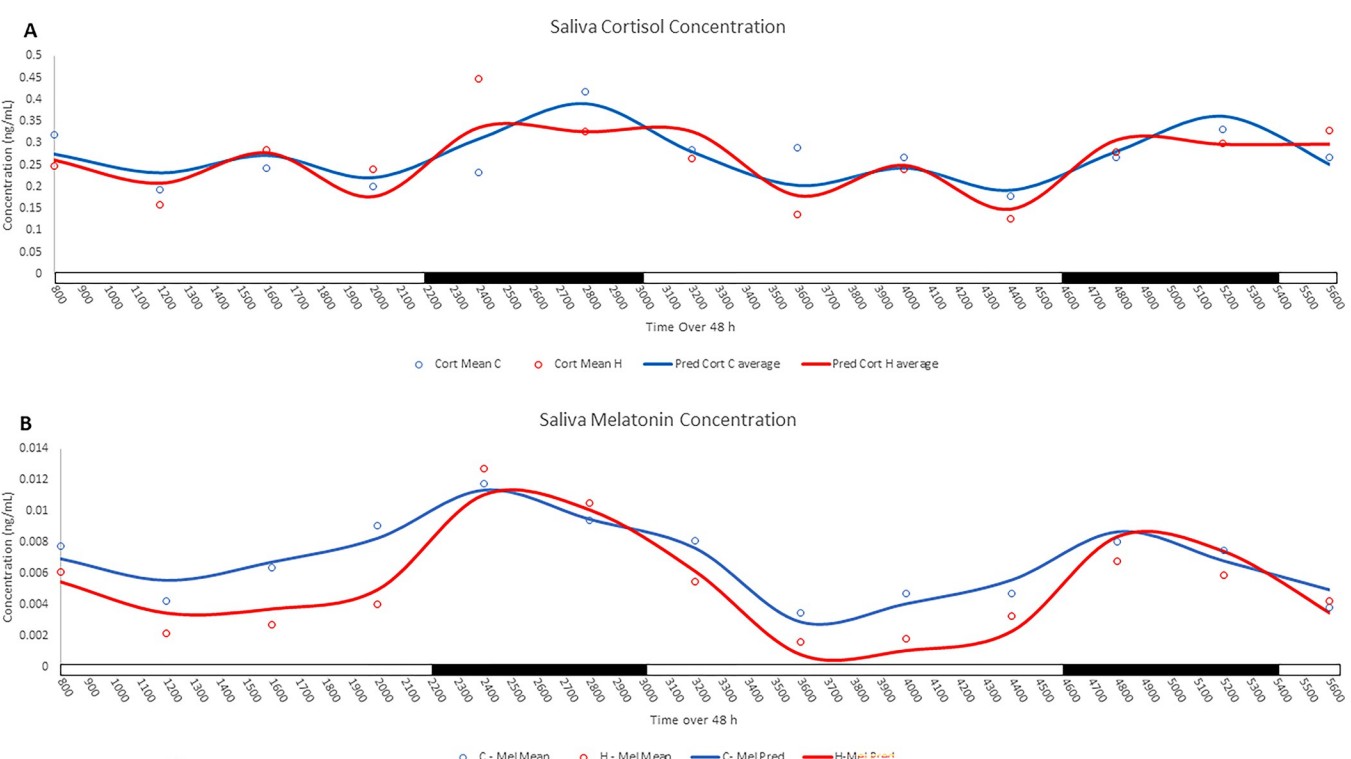

**Fig 5. Effect of cooling on circadian rhythms of salivary cortisol and melatonin concentration.** Effect of cooling (blue) or H (red) on sow salivary cortisol (A) and melatonin (B) concentrations over 48 h and the fitting to sine and cosine curves. The black bar on the x-axis represents the dark phase, while the white bar represents the light phase.

There was a significant treatment by sine single phase variable interaction (P < 0.05), indicating a difference in the 24 h rhythm.

## Cooling effect on relative fold change of saliva cytokines

Levels of interleukin (IL) IL-1 beta, IL-10, IL-15, and transforming growth factor (TGF) beta 1 varied in response to treatments (Fig 6). IL-1 beta and TGF-beta 1 showed higher diurnal fold changes in H sows compared to C sows, with IL-1 beta being higher at 2000 (pm) and TGF-beta 1 higher at 0800 (am). On the other hand, IL-10 and IL-15 had higher fold changes between 0800 (am) and 2000 (pm) in C sows compared to H sows, with IL-10 higher 0800 (am) and IL-15 higher at 2000 (pm). Both treatments had similar diurnal patterns for IL-1a and IL-1 beta, with higher levels at 2000 (pm) and for TGF beta 1 with higher levels in the 0800 (am). Cytokines not shown are below detection.

## Discussion

The recent revelation that the core clock genes ARNTL (BMAL1) and NPAS2 were linked to HS response in multiple species [28], lead to our present investigation into whether cooling lactating sows exposed to natural heat waves affected circadian patterns of behavior and physiology. Cooling strategies mitigate some of the negative effects of HS [8, 19–21], and since circadian clock genes are related to HS response, we hypothesized that circadian rhythms of behavior and primary outputs of the SCN (core body temperature and cortisol and melatonin circadian rhythms) would be altered by cooling. The primary finding of this study was that cooling lactating sows during a 48 h period of an early summer heat wave decreased overall

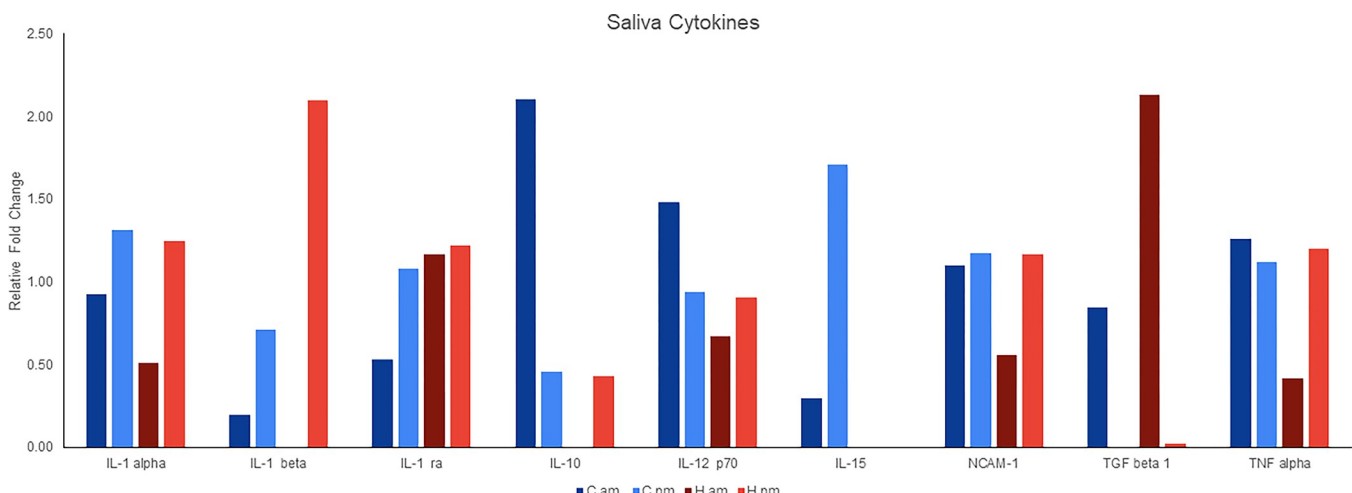

**Fig 6. Impact of cooling on diurnal salivary cytokine levels.** The effect of cooling (blue) HS (red) sows on the relative abundance of cytokines. Levels are relative to average densitometry of positive control spots on the array. IL-1a_Interleukin 1 alpha; IL-1b_Interleukin 1 beta; IL-1Ra_Interleukin 1 receptor antagonist; IL-10_Interleukin 10; IL-12 p70_Interleukin 12; IL-15_Interleukin 15; NCAM-1_Neural cell adhesion molecule 1; TGF beta 1_Transforming growth factor beta 1; TNF alpha 1_ Tumor necrosis factor alpha 1.

daily RR and RT and increased melatonin levels, with the novel discovery of alterations in circadian rhythms of melatonin in saliva, and circadian patterns of eating which related to variations in daily patterns of core body temperature. However, contrary to our hypothesis, cooling had no effect on salivary cortisol concentrations or rhythms. In addition, the analysis of saliva cytokine levels within treatment by timepoint demonstrated that similar to blood, inflammation status biomarkers in saliva could reflect physiological responses to heat stress. Fig 7 shows the impact of cooling on circadian patterns of the multiple variables measured in sows exposed to early summer heat stress conditions, with potential mechanisms of action proposed.

The cooling treatment reduced overall RR and RT of sows. The fitted curve of RR displayed a circadian rhythm in both C and H sows, with distinct differences in breaths per minute over the 48 h study period. The mean RR of cooled sows over the 48 h period was 53.5 breaths per minute lower than heat-stressed sows, and the RR was significantly higher on day 1, with an estimate of 8.26 breaths per minute more than on day 2. Cooling lowered body temperature across the entire 48 h period by about 0.4˚C compared to the heat-stressed sow group. The cooling effects of the pad in reducing RR and maintaining a narrow range of RT were consistent with previous reports [21, 24].

The periodic regression model revealed RR exhibited sine-cosine rhythms every 24 and 8 h in both groups. Whereas RT displayed sine-cosine rhythms every 24, 12, and 8 h. The fitted RT curve reflected a circadian rhythm in both C and H sows. However, the 8 h rhythm, most prominent in H sows, resulted in two peaks in temperature during the dark phase. The Fourier model indicated that the second peak in the dark phase can likely be explained by the increase in body temperature in response to the heat of metabolism from feed consumption [29]. The H sows shifted their phase of eating relative to C sows. Peak of eating activity of cooled sows was between 1500 and 1600, when room temperature was greatest. Whereas H sows exhibited almost no eating behavior at 1500 and 1600, and delayed eating activity to 2000 to 2100 when ambient-room temperatures were cooler. These findings are consistent with previous observations [29] of HS sows delaying feed intake to cooler hours of the day. The second peak in RT in H sows during the dark phase is significant in terms of an animal welfare perspective and based on this observation it may be important to continue cooling fans and other means to

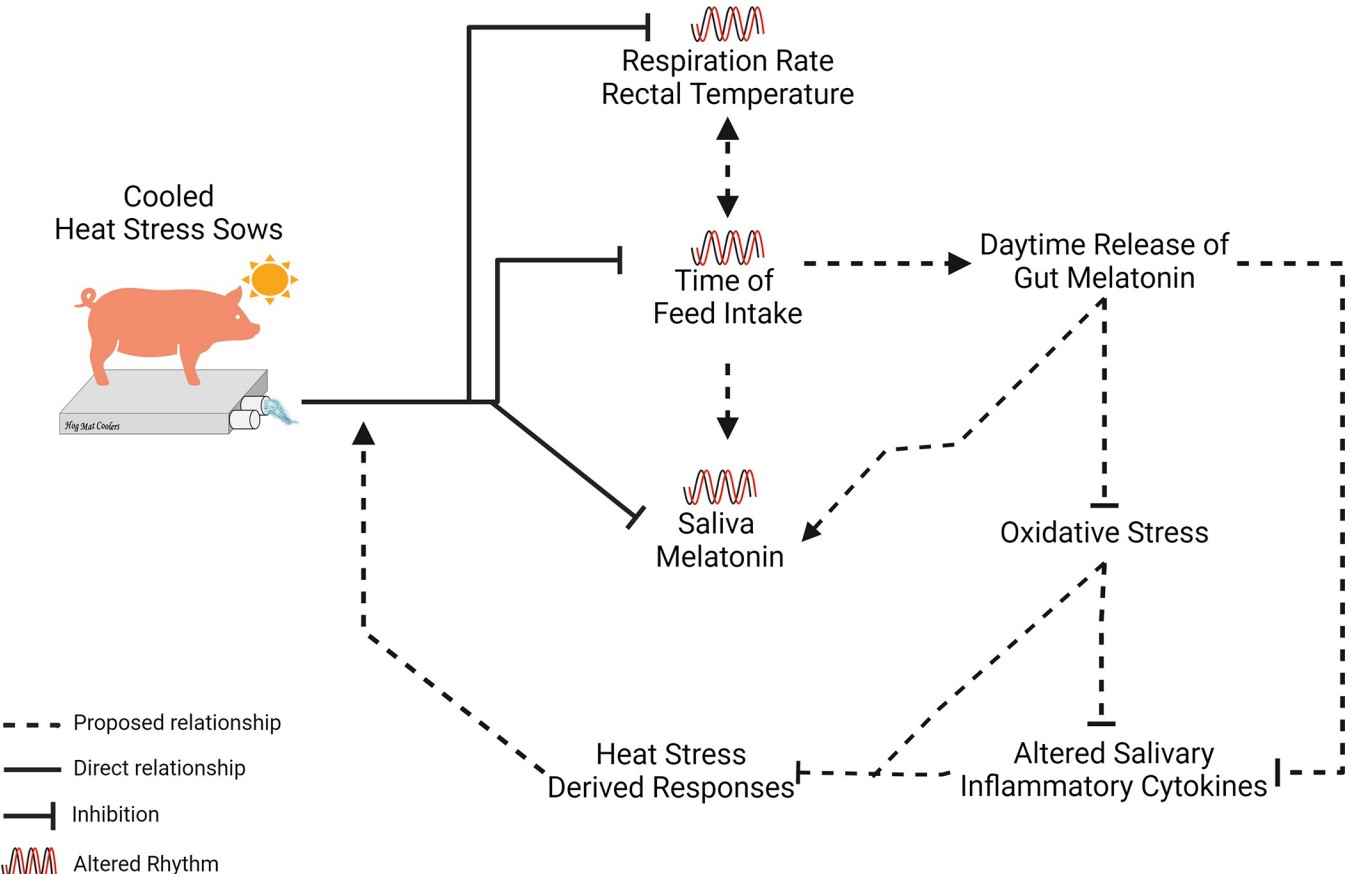

**Fig 7. Schematic of proposed mechanism of cooling ameliorations of heat stress.** Cooling represses alterations in daily patterns of respiration rate, rectal temperature, timing of feed intake, and salivary melatonin concentrations. We propose that cooling reduces the shift in feeding time, which likely increases daytime melatonin release from the gut, raising daytime salivary melatonin levels and decreasing oxidative stress and salivary inflammatory cytokines. These effects mitigate negative heat stress responses, thereby supporting the positive impact of cooling on circadian rhythms. The figure also suggests that changes in feeding time can influence rectal temperature and vice versa.

mitigate effects of environmental temperatures, even after ambient temperature drops. Although not measured here, others reported heat stress also alters eating behavior by reducing the total amount consumed by sows and rate of feed consumption [30]. Behavior is an input of the circadian timing system [31], and timing of feed intake serves as a primary temporal input to peripheral clocks, thus the alterations in eating behavior to later and cooler parts of the day in response to high heat disrupt the normal regular activity of the sow, which in turn can disrupt physiological rhythms.

Although cooling did not affect the overall time the sows engaged in any activity over the 48 h period, it affected the time certain activities were performed during the day including eating, standing, and laying. Cooling increased eating behavior in cooled sows during the day, irrespective of the peak of heat, and this eating behavior coincided with standing behavior as sows most often stand up to eat. Laying, on the other hand, is an activity inversely related to standing and sitting. However, considering that heat produced from standing after feeding doubles in comparison to the heat produced from laying down [32], the standing and laying behaviors of the HS sows could be reflective of an entrained response to HS in order to reduce heat production generated from these activities.

Nursing behavior varied by time of day, with a tendency for increased nursing activity during the light phase and lower nursing activity in the dark phase. Active nursing behavior was not influenced by the cooling treatment, indicating sows allowed their litter to nurse at least once or twice every hour, regardless of the ambient HS. It is widely understood that milk production is associated with litter demand [33], and so findings from this study indicate that the lower milk production resulting from HS [8] may not be associated with a decreased demand of milk or decreased removal of milk from glands by neonates, as the sows in this study did not deny the piglets from active nursing behavior.

Saliva melatonin levels exhibited a circadian rhythm in both H and C sows, with levels peaking in the dark phase. This pattern of melatonin is expected as the synthesis of melatonin in the pineal gland is inhibited by light information coming from the SCN [34, 35]. Although the timing of the acrophase was similar between C and H sows, cooling sows led to higher melatonin concentrations across the entire study, with elevated levels evident in the light phase of the cycle. In addition to being produced in the pineal gland melatonin is also produced in high quantities by the enterochromaffin cells of the gastrointestinal tract (GIT) [36] and has been shown to be secreted by serous cells in the parotid of the salivary gland [37]. The levels produced by the GIT are greater than the pineal gland and contribute to circulating concentrations of melatonin, especially during the daytime when melatonin synthesis is induced by feeding [38, 39]. Although we did not measure feeding activity in the second phase of the study, we presume that the greater feeding activity during the light phase in the cooled sows versus the later eating activity in the HS group closer to the onset of the dark phase, contributed to the higher levels of saliva melatonin in cooled sows potentially from the enterochromaffin cells in the GIT or cells within salivary gland. Melatonin is a potent antioxidant, actively scavenging free radicals and exerting anti-inflammatory effects directly or indirectly through endocrine and paracrine activity [40]. The higher level of melatonin in cooled sows may be a potential mechanism by which the negative effects of HS are mitigated in these animals. A study by Tian et. al. (2013) on heat stressed mice found supplementing animals with melatonin mitigates heat stress effects on hypothalamus by increasing levels of anti-inflammatory cytokines, reducing pro-inflammatory cytokines and reducing oxidative damage, indicating a link between melatonin and HS and its mitigative role [41]. In contrast to our initial hypothesis, the level of saliva cortisol was not different between cooled and HS groups. These results were unexpected as others have reported increased cortisol [42]. However, our results align with findings of others [43], which reported no difference in hair cortisol levels between chronic heat-stressed sows provided with cooling pads and those provided with low water drinking temperature. These findings reflect variable responses in cortisol release of sows exposed to heat stress and indicate that levels may not be a reliable marker for assessment of animal physiology-welfare in regards to response to high ambient heat and humidity. This interpretation is supported by a recent systematic review that reported cortisol levels in pigs may be more indicative of stressors like social or immune stress rather than of heat stress [44].

It is worth noting that saliva sampling as opposed to conventional blood sampling has gained interest in pigs due to it being less stressful and relatively non-invasive approach providing better welfare for animals and more comfort for human staff involved in collection. Use of saliva to screen for physiological and pathological status of the animal is supported by a growing abundance of data indicating that saliva can be used to screen for biomarkers of stress, inflammation, and immune response [45–47]. To this end, we also investigated whether there were differences in cytokines between the treatments as well as time of day using pooled saliva samples. We found that cooling modified IL-10 and IL-1 beta levels, with IL-10, 1.5-fold lower in HS versus cooled sows at 800 (AM) and IL-1 beta, 1.5-fold higher in H than C sows at 2000 (PM). The alterations of these cytokines in response to heat stress are consistent with research

in rodents [48], which reported HS lowered IL-10 levels and increased IL-1 beta in various organs. The reduced level of IL-10 in the H compared to C group could indicate a delay or impairment in the anti-inflammatory response in the heat-stressed animals as IL-10 is known for its potent anti-inflammatory properties and its role in controlling inflammatory responses [49, 50]. Such a decrease in IL-10 levels could as well promote pro-inflammatory cytokine IL-1 beta [49]. Interestingly, IL-1 beta appears to be a more accurate indicator of systemic inflammation than IL-1 alpha [51] which could explain why its greater response relative to IL-1 alpha in the saliva of heat-stressed sows. Additionally, IL-1 beta has been found to impact the expression of PER2 and BMAL1 (ARNTL) components of the circadian clock in human chondrocytes [52], which suggests that alterations in its expression might impact the circadian rhythm in the heat-stressed sows. In view of IL-10's possible impact on circadian clocks, its low level in HS versus C sows could also suggest a potential disruption in their circadian rhythm, as a study of humans found lower IL-10 levels in shift workers with circadian misalignment than the control group [15].

TGF-beta 1 levels showed diurnal variation in both H and C sow saliva, with levels undetectable in the AM sample, and high in the PM sample. Levels of TGF-beta 1 were greater in saliva of H versus C. TGF-beta 1 is denatured and reformed during heat increment and heat reduction, respectively [53]. The high level of TGF-beta 1 in the morning and its subsequent reduction during the day might be influenced by the fluctuating air temperature. TGF-beta 1 is known for its pleiotropic effects, showing both pro-inflammatory and antioxidant properties [54], and associated with the regulation of circadian rhythms, influencing the expression of positive and negative oscillators of the circadian clock. Thus, the differences in H and C animals may affect circadian clocks and warrant further investigation. Saliva IL-15 also showed diurnal variation in the cooled sows, being greater in PM than AM. However, IL-15 was undetectable in the H sows. Studies of humans found elevated IL-15 in saliva related to disease and inflammation [55], which would predict the opposite to our finding. However, Silva and others [56], found IL-15 to play an anti-inflammatory role. As such, the lower expression of IL-15 in the saliva of the heat-stressed group compared to the cooled sows could be indicative of its anti-inflammatory role being overpowered by the predominant expression of pro-inflammatory cytokines like IL-1 beta. Despite these intriguing findings, not much is currently known about the relationship between IL-15 and circadian patterns. Still, it is interesting to further note that the elevated concentrations of melatonin during the light phase may have contributed to increased IL-10 levels observed in cooled sows, supporting the anti-inflammatory effects of melatonin. This is consistent with findings by Tian et al. (2013), who reported that melatonin treatment enhanced IL-10 concentrations and reduced levels of IL-1 beta in the hypothalamus of heat-stressed mice [41]. Together data represents an exploratory analysis of whether biomarkers of inflammation often related to heat stress could be detected in saliva and whether they varied by time of day. Although pooling samples decreased the power of the study, data demonstrate markers of inflammation in the saliva that can help direct future research in this area.

In conclusion, cooling of sows during early summer HS conditions reduced RR and RT and altered their daily rhythms. Cooling also altered timing of sow standing, laying, and eating behaviors, but had no significant effect on active nursing. Relative to cooled sows, HS sow eating activity was delayed to cooler times of the day and related to alterations in the daily rectal temperature rhythm, including additional increments in temperature. While cooling did not affect saliva cortisol rhythms or levels, saliva melatonin levels were greater in the cooled sows across the circadian cycle. Together these data indicate that initial summer HS alters circadian rhythms of physiology and behavior and may potentially be considered a chronodisruptor. Understanding the manner that HS disrupts circadian clock functions is likely important to

animal production, as melatonin is a potent anti-oxidant and regulation of temperature rhythms is key to synchronizing physiological systems across the body including metabolism and the endocrine system which are key regulators of lactation. Thus, these data demonstrate that circadian pattern of behavior, physiology and hormones are altered by cooling HS sows and provide support for further investigation into the inter-relationship between circadian clocks and HS response, as well as using saliva samples in biomarker determination of HS response.

## Supporting information

**S1 Table. Dataset for created and shared figures.** Room conditions, sow health record, sow behavior record, eating behavior and relationship with rectal temperature, salivary hormone concentrations, salivary cytokine densitometry.
(XLSX)

## Acknowledgments

The authors would like to the Purdue Swine Unit staff for enabling the field experiment to be conducted. We would also like to thank members of the Minor Laboratory of Animal Immune Response Lab and Casey Mammary Gland Biology Lab for assisting in the analysis.

## Author Contributions

**Conceptualization:** Allan Schinckel, Radiah Minor, Theresa Casey.

**Data curation:** Wonders Ogundare, Kelsey Teeple, Elizabeth Fisher, Corrin Davis, Leriana Garcia Reis, Amber Jannasch, Linda M. Beckett, Allan Schinckel, Theresa Casey.

**Formal analysis:** Wonders Ogundare, Corrin Davis, Leriana Garcia Reis, Amber Jannasch, Allan Schinckel.

**Funding acquisition:** Radiah Minor, Theresa Casey.

**Investigation:** Wonders Ogundare, Kelsey Teeple, Elizabeth Fisher, Linda M. Beckett, Theresa Casey.

**Methodology:** Theresa Casey.

**Project administration:** Kelsey Teeple, Theresa Casey.

**Supervision:** Allan Schinckel, Theresa Casey.

**Writing – original draft:** Wonders Ogundare.

**Writing – review & editing:** Leriana Garcia Reis, Amber Jannasch, Linda M. Beckett, Allan Schinckel, Radiah Minor, Theresa Casey.

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
