## [Decision Letter · Decision Letter 0]

30 Apr 2024

PONE-D-23-42577Cooling lactating sows exposed to acute summer heat wave alters circadian patterns of behavior and rhythms of respiration, rectal temperature, and saliva melatoninPLOS ONE

Dear Dr. Casey,

Thank you for submitting your manuscript to PLOS ONE. After careful consideration, we feel that it has merit but does not fully meet PLOS ONE’s publication criteria as it currently stands. Therefore, we invite you to submit a revised version of the manuscript that addresses the points raised during the review process.

I would like to sincerely apologise for the delay you have incurred with your submission. It has been exceptionally difficult to secure reviewers to evaluate your study. We have now received three completed reviews; the comments are available below. The reviewers have raised significant scientific concerns about the study that need to be addressed in a revision.

Please revise the manuscript to address all the reviewer's comments in a point-by-point response in order to ensure it is meeting the journal's publication criteria. Please note that the revised manuscript will need to undergo further review, we thus cannot at this point anticipate the outcome of the evaluation process.

We look forward to receiving your revised manuscript.

Kind regards,

Miquel Vall-llosera Camps

Staff Editor

PLOS ONE

 [This research was supported by the U.S. Department of Agriculture, National Institute of Food and Agriculture (USDA-NIFA) grant 2021-07097 The impact of moringa on milk quality and quantity and piglet health in heat stressed swine. ].  

3. We note that your Data Availability Statement is currently as follows: [All relevant data are within the manuscript and its Supporting Information files]

Reviewers' comments:

Reviewer's Responses to Questions

**Comments to the Author**

1. Is the manuscript technically sound, and do the data support the conclusions?

Reviewer #1: Partly

Reviewer #2: Yes

Reviewer #3: Yes

2. Has the statistical analysis been performed appropriately and rigorously? 

Reviewer #1: N/A

Reviewer #2: Yes

Reviewer #3: Yes

3. Have the authors made all data underlying the findings in their manuscript fully available?

Reviewer #1: Yes

Reviewer #2: Yes

Reviewer #3: Yes

4. Is the manuscript presented in an intelligible fashion and written in standard English?

Reviewer #1: Yes

Reviewer #2: Yes

Reviewer #3: Yes

5. Review Comments to the Author

Reviewer #1: This MS was poorly designed to measure the effect of HS on early lactating sows. In order to achieve HS environmental conditions should be controlled and approve the existence of chronic or acute HS in the animals by physiological parameters. The cytokines saliva measurements are not applicable. During overall the MS there is not clear idea of Pro or anti-inflammatory reaction of the immune system to keep hemostasis.

Line comment

48 The word inflammation is mentioned in the reference only once and its results of complications...

"Consequently, the gastrointestinal tract vasoconstricts in an effort to support the altered blood distribution and the reduced splanchnic blood and nutrient flow creates intestinal barrier dysfunction. Intestinal infiltrating antigens stimulate a local immune reaction and, if severe enough, cause systemic endotoxemia associated with inflammation and an acute phase protein response".

107 What NRC ? Ver etc…..

108 Add a table of the composition of the diet and ingredients.

135 during all 48 h ?

148 this does not apply for the RT measurements every 60 min

169 record g forces and not rpm

243 It is 0400h ?

244 3100?

322 Since data was obtained by only 1 sample per treatment, you actually can not say anything!

423 this references, do not refer cytokine analysis via saliva.

Reviewer #2: The research manuscript titled-"Cooling lactating sows exposed to acute summer heat wave alters circadian patterns of behavior and rhythms of respiration, rectal temperature, and saliva melatonin" submitted by Theresa Casey investigated and reported interesting findings which will effectively improve the Pig production system in minimizing the impact of heat stress. The study also reported the significance of cooling in lactating sows and how it impacted the circadian rhythm and its associated physiology. Heat stress and its associated impact on animal production and health is well known. The impeding climate change is further going to be worse in coming time and therefore to maintain the pace in animal production system suitable mitigating strategies are warranted. The study is well planned and executed and number of novel findings are proposed by the authors. The study results concluded that circadian pattern of behavior, physiology and hormones are altered by cooling heat stress sows. I can see another potential finding of the study which will be used as a biomarker (salivary melatonin) and will be used as a biomarker for HS detection. I am highly convinced with the findings of the study; however, before reaching to any final conclusion, I have few suggestions for the authors to consider in this present form of the manuscript during their revision and submission to the journal.

1. Shorten the introduction and frame a clear hypothesis basing on the background of the study.

2. Please mention the studies related or already done in the similar line in pigs or in any other mammalian system.

3. Try to establish a link between the heat stress and melatonin system.

4. Is it possible to propose a figure or a mechanistic picture out of the findings displaying the best possible interplay between the heat stress and melatonin system for better clarity and effective scientific understanding?

5. How cooling is altering the circadian rhythms need to be discussed with relevant literature or more emphasis on this?

6. Why salivary cortisol levels are not altered in HS sows?

7. Is there any relationship between lactation stress and heat stress or combinedly both affecting the production needs to be discussed.

8. How immunological parameters were improved due to cooling in HS sows need to be discussed in the discussion with probable literature.

9. Do the authors think that melatonin and circadian rhythm are the major factors determining the effects of cooling on heat stress sows in improving their physiological parameters, and if so how that is possible can be explained more in the discussion?

10. Whether the authors measured Prolactin and blood metabolite parameters like NEFA, lipid profile, BUN, blood glucose levels and other hormones like insulin and thyroxine. If so can it be provided here and if not it is suggested to do. I was also interested to see the heat shock proteins and their effective role after cooling.

best wishes

Reviewer #3: This manuscript is well designed and organized and has scientific sound at the applicable level in swine industry. I really enjoyed reading this manuscript. However, I just have minor points as follows:

The sample size calculation is not provided, please provide your rationale for not performing a sample size calculation in the M&M section.

I would recommend including a figure in the M&M section to describe temperature-humidity index throughout the experiment.

6. PLOS authors have the option to publish the peer review history of their article (what does this mean?). If published, this will include your full peer review and any attached files.

Reviewer #1: No

Reviewer #2: **Yes: **DILIP K SWAIN

Reviewer #3: No

---

## [Author Response · Author response to Decision Letter 0]

14 Jun 2024

Dear Associate Editor and Reviewers,

Thank you for the thorough review of our manuscript, addressing your comments improved the quality of the manuscript. Here you will find: 

• A rebuttal letter that responds to each point raised by the academic editor and reviewer(s). 

• A marked-up copy of your manuscript that highlights changes in red font made to the original version.

• An unmarked version of your revised paper without tracked changes.

The following is the financial disclosure statement.

This research was supported in part by the U.S. Department of Agriculture, National Institute of Food and Agriculture, award number, 2022-67016-36194, titled: Using cooling & moringa oleifera to mitigate oxidative stress and inflammation in heat stressed sows & improve milk production and performance. The findings and conclusions in this publication have not been formally disseminated by the U. S. Department of Agriculture and should not be construed to represent any agency determination or policy. 

The cooling pads were provided by the IHT group, Oak Bluff, Manitoba, Canada. 

The funders and IHT had no role in study design, data collection and analysis, decision to publish, or preparation of the manuscript.

3. We note that your Data Availability Statement is currently as follows: [All relevant data are within the manuscript and its Supporting Information files]

Response. We uploaded all data as Supporting Information files 

Reviewer #1: 

1. This MS was poorly designed to measure the effect of HS on early lactating sows. In order to achieve HS environmental conditions should be controlled and approve the existence of chronic or acute HS in the animals by physiological parameters. 

Response: We respectfully disagree with the reviewer. Applying a cooling treatment in a hot-humid climate is an accepted experimental treatment (e.g. PMID: 32113780) to understand the impact of HS on livestock physiology. In regards to this particular study, the thermoneutral zone of lactating sows is reported to be 18-22 C, with panting becoming evident ~25C (https://doi.org/10.1017/S1751731111002448). The elevated rectal temperatures and respiration rate indicated that H relative to C animals were experiencing a heat stress response. Whereas the cooling pad treatment maintained animals in thermoneutral conditions through conductive cooling mechanisms as evident by low RR and normal RT, despite high ambient heat and humidity. We defined the HS exposure of the sows as acute because it was the first heat wave the animals experienced that summer period. 

2. The cytokines saliva measurements are not applicable. During overall the MS there is not clear idea of Pro or anti-inflammatory reaction of the immune system to keep hemostasis. 

Response: The data represents an exploratory analysis of whether biomarkers of inflammation often related to heat stress could be detected in saliva and whether they varied by time of day. Although pooling samples decreases the power of the study, as we now highlight in our discussion, data demonstrate markers of inflammation in the saliva that can help direct future research in this area.

Line comment

48 The word inflammation is mentioned in the reference only once and its results of complications...

"Consequently, the gastrointestinal tract vasoconstricts in an effort to support the altered blood distribution and the reduced splanchnic blood and nutrient flow creates intestinal barrier dysfunction. Intestinal infiltrating antigens stimulate a local immune reaction and, if severe enough, cause systemic endotoxemia associated with inflammation and an acute phase protein response". 

Response: We added other references that gives more information on heat stress impact on immune system and inflammatory response:

5. Chen S, Yong Y, Ju X. Effect of heat stress on growth and production performance of livestock and poultry: Mechanism to prevention. J Therm Biol. 2021;99:103019. Epub 2021/08/24. doi: 10.1016/j.jtherbio.2021.103019. PubMed PMID: 34420644.

6. Mayorga EJ, Renaudeau D, Ramirez BC, Ross JW, Baumgard LH. Heat stress adaptations in pigs. Animal Frontiers. 2018;9(1):54-61. doi: 10.1093/af/vfy035.

7. Pearce SC, Mani V, Boddicker RL, Johnson JS, Weber TE, Ross JW, et al. Heat stress reduces barrier function and alters intestinal metabolism in growing pigs. J Anim Sci. 2012;90 Suppl 4:257-9. Epub 2013/02/13. doi: 10.2527/jas.52339. PubMed PMID: 23365348.

107 What NRC ? Ver etc…..

Response: It’s NRC 2012, added.

108 Add a table of the composition of the diet and ingredients.

Response: Table of composition has been added.

135 during all 48 h ?

Response: Yes. 

148 this does not apply for the RT measurements every 60 min

Response: Indeed RT measurement could have conflicted with other data collection. 

169 record g forces and not rpm

Response: rpm has been converted to g force. 

243 It is 0400h ? 

Response: It is 4000 which is 1600 (4pm) on the second day.

244 3100? 

Response: We would like to leave the axis annotation as it is with a continuous time count of the 48 hour period.

322 Since data was obtained by only 1 sample per treatment, you actually can not say anything!

Response: We analyzed a pool of saliva of all the animals within each treatment by timepoint, which although it precludes statistical analysis, the changes in the differences in the factors highlighted represent the population as a whole. Importantly, this exploratory analysis demonstrates that saliva, and not just blood, may reflect heat stress induced changes in immune response and inflammation.

As indicated above we addressed this as a limitation in our discussion.

423 this references, do not refer cytokine analysis via saliva. 

Response: The references refer broadly to saliva being explored as biomarker of inflammation and does not refer to cytokines in particular. However, another reference has been added that refers to stress evaluations of cytokines in saliva. 

Reviewer #2: The research manuscript titled-"Cooling lactating sows exposed to acute summer heat wave alters circadian patterns of behavior and rhythms of respiration, rectal temperature, and saliva melatonin" submitted by Theresa Casey investigated and reported interesting findings which will effectively improve the Pig production system in minimizing the impact of heat stress. The study also reported the significance of cooling in lactating sows and how it impacted the circadian rhythm and its associated physiology. Heat stress and its associated impact on animal production and health is well known. The impeding climate change is further going to be worse in coming time and therefore to maintain the pace in animal production system suitable mitigating strategies are warranted. The study is well planned and executed and number of novel findings are proposed by the authors. The study results concluded that circadian pattern of behavior, physiology and hormones are altered by cooling heat stress sows. I can see another potential finding of the study which will be used as a biomarker (salivary melatonin) and will be used as a biomarker for HS detection. I am highly convinced with the findings of the study; however, before reaching to any final conclusion, I have few suggestions for the authors to consider in this present form of the manuscript during their revision and submission to the journal.

1. Shorten the introduction and frame a clear hypothesis basing on the background of the study.

Response: The introduction is further shortened and hypothesis more clearly defined

2. Please mention the studies related or already done in the similar line in pigs or in any other mammalian system.

Response: Similar studies on circadian disruptions in heat stress on cattle is now briefly mentioned in the introduction. 

3. Try to establish a link between the heat stress and melatonin system.

Response: In our discussion section we present our hypothesis that heat stress reduces feeding during the day and subsequently melatonin production from GIT enterochromaffin cells and added Fig. 7 schematic to propose the potential mechanistic relationships.

4. Is it possible to propose a figure or a mechanistic picture out of the findings displaying the best possible interplay between the heat stress and melatonin system for better clarity and effective scientific understanding?

Response: see response to number 3.

5. How cooling is altering the circadian rhythms need to be discussed with relevant literature or more emphasis on this?

Response: By adding the figure as proposed, we believe we addressed this concern, as well as adding Tian et al. reference.

6. Why salivary cortisol levels are not altered in HS sows?

Response: We added a discussion indicating that a systematic review of data found that cortisol levels may be more indicative of other stressors like social or immune stress than of heat stress. (Guevara RD et al.)

7. Is there any relationship between lactation stress and heat stress or combinedly both affecting the production needs to be discussed.

Response: Yes. Lactation is accompanied by high load of metabolic heat from milk production which is compounded by high external temperatures and impairs sow productivity, which is now discussed in the introduction.

8. How immunological parameters were improved due to cooling in HS sows need to be discussed in the discussion with probable literature.

Response We believe that melatonin anti-inflammatory action may have contributed to modulate cytokine levels in cooled sows and is now discussed, and presented in Fig. 7.

9. Do the authors think that melatonin and circadian rhythm are the major factors determining the effects of cooling on heat stress sows in improving their physiological parameters, and if so how that is possible can be explained more in the discussion?

Response: We think that circadian rhythms and melatonin are factors influenced by the cooling effect to mitigate the negative impact of heat stress in lactating sows.

10. Whether the authors measured Prolactin and blood metabolite parameters like NEFA, lipid profile, BUN, blood glucose levels and other hormones like insulin and thyroxine. If so can it be provided here and if not it is suggested to do. I was also interested to see the heat shock proteins and their effective role after cooling. 

Response: We agree it would have been good to look at these as they are factors already known but measuring them is beyond the scope of this study, which aimed to explore circadian patterns and we avoided blood sampling as the means of restraint typically required increases stress of swine.

Reviewer #3: This manuscript is well designed and organized and has scientific sound at the applicable level in swine industry. I really enjoyed reading this manuscript. However, I just have minor points as follows:

A. The sample size calculation is not provided, please provide your rationale for not performing a sample size calculation in the M&M section. 

Response: The priori sample size calculations indicated we need 12 animals in each treatment group but multiple sows were lost due to diseases unrelated to the study. Regardless, we decided to continue on with the project that was highly exploratory. Power analysis was conducted for each of the sample sizes proposed. With 10 animals per treatment, and a 1.5-fold difference in variable with 0.5 standard deviation, the power of the study is 0.80; if the difference increased 1.75, then a sample size of 4 was needed for a power of 0.8. With n=9 animals in each group, and a 1.75 difference, with 0.5 standard deviation, then the power of the study is 0.9.

B. I would recommend including a figure in the M&M section to describe temperature-humidity index throughout the experiment. 

Response: We provide the temperature, relative humidity and dew points (supplemental information) for the sows that can be used by others to calculate temperature humidity index although involves complex calculations, but there is no temperature humidity index up to date for modern lactating sows.

---

## [Decision Letter · Decision Letter 1]

31 Jul 2024

Cooling lactating sows exposed to early summer heat wave alters circadian patterns of behavior and rhythms of respiration, rectal temperature, and saliva melatonin

PONE-D-23-42577R1

Dear Dr. Casey,

We’re pleased to inform you that your manuscript has been judged scientifically suitable for publication and will be formally accepted for publication once it meets all outstanding technical requirements.

Kind regards,

Andras Garami, M.D., Ph.D.

Academic Editor

PLOS ONE

Additional Editor Comments (optional):

Reviewers' comments:

Reviewer's Responses to Questions

**Comments to the Author**

1. If the authors have adequately addressed your comments raised in a previous round of review and you feel that this manuscript is now acceptable for publication, you may indicate that here to bypass the “Comments to the Author” section, enter your conflict of interest statement in the “Confidential to Editor” section, and submit your "Accept" recommendation.

Reviewer #2: All comments have been addressed

Reviewer #3: All comments have been addressed

2. Is the manuscript technically sound, and do the data support the conclusions?

Reviewer #2: Yes

Reviewer #3: Yes

3. Has the statistical analysis been performed appropriately and rigorously? 

Reviewer #2: Yes

Reviewer #3: Yes

4. Have the authors made all data underlying the findings in their manuscript fully available?

Reviewer #2: Yes

Reviewer #3: Yes

5. Is the manuscript presented in an intelligible fashion and written in standard English?

Reviewer #2: Yes

Reviewer #3: Yes

6. Review Comments to the Author

Reviewer #2: (No Response)

Reviewer #3: (No Response)

7. PLOS authors have the option to publish the peer review history of their article (what does this mean?). If published, this will include your full peer review and any attached files.

Reviewer #2: **Yes: **DILIP K SWAIN

Reviewer #3: No

---

## [Editor Report · Acceptance letter]

12 Sep 2024

PONE-D-23-42577R1 

PLOS ONE

Dear Dr. Casey, 

I'm pleased to inform you that your manuscript has been deemed suitable for publication in PLOS ONE. Congratulations! Your manuscript is now being handed over to our production team.

Kind regards, 

on behalf of

Dr. Andras Garami 

Academic Editor

PLOS ONE